# Taking It Personally: 3D Bioprinting a Patient-Specific Cardiac Patch for the Treatment of Heart Failure

**DOI:** 10.3390/bioengineering9030093

**Published:** 2022-02-25

**Authors:** Niina Matthews, Berto Pandolfo, Daniel Moses, Carmine Gentile

**Affiliations:** 1School of Biomedical Engineering, Faculty of Engineering and IT, University of Technology Sydney (UTS), Sydney, NSW 2007, Australia; niina.matthews@student.uts.edu.au; 2School of Design, Faculty of Design, Architecture and Building, University of Technology Sydney (UTS), Sydney, NSW 2007, Australia; berto.pandolfo@uts.edu.au; 3Research Imaging NSW, University of New South Wales (UNSW), Sydney, NSW 2034, Australia; daniel.moses@unsw.edu.au; 4Sydney Medical School, Faculty of Medicine and Health, The University of Sydney, Sydney, NSW 2000, Australia

**Keywords:** 3D bioprinting, patient-specific, cardiac patch, heart failure, 3D modelling

## Abstract

Despite a massive global preventative effort, heart failure remains the major cause of death globally. The number of patients requiring a heart transplant, the eventual last treatment option, far outnumbers the available donor hearts, leaving many to deteriorate or die on the transplant waiting list. Treating heart failure by transplanting a 3D bioprinted patient-specific cardiac patch to the infarcted region on the myocardium has been investigated as a potential future treatment. To date, several studies have created cardiac patches using 3D bioprinting; however, testing the concept is still at a pre-clinical stage. A handful of clinical studies have been conducted. However, moving from animal studies to human trials will require an increase in research in this area. This review covers key elements to the design of a patient-specific cardiac patch, divided into general areas of biological design and 3D modelling. It will make recommendations on incorporating anatomical considerations and high-definition motion data into the process of 3D-bioprinting a patient-specific cardiac patch.

## 1. Introduction

Ischaemic heart disease persists as the main cause of death globally, accounting for 16% of all deaths in the world [1]. Although the age-standardised rate of global mortality due to cardiovascular disease (CVD) declined by 27% between 2000 and 2019, population growth and ageing increased the total number of deaths from ischaemic heart disease by 1.8 million at the same time [2]. Ischaemic heart disease, or coronary artery disease (CAD), blocks the coronary arterial circulation, which leads to cardiomyocyte death in the ischemic region [3]. Chronic CAD can cause a myocardial infarction (MI) event when one or more coronary arteries are obstructed. Cardiac remodelling causes post-infarct fibrotic scarring, stiffening of the myocardial wall and death of cardiomyocytes and can eventually result in a heart failure (HF) [4]

Patients with HF suffer from a poor quality of life and have an average 5-year survival rate of 60% [5]. As the condition is progressive and irreversible, the gold-standard treatment in end-state HF is a heart transplant [4,6]. This treatment option is not accessible for everyone as there is a wide gap between the number of transplant donors and potential recipients. Each year, less than 6000 heart transplants are performed globally, which is in stark contrast to the estimated 26 million people living with HF [7,8]. A heart transplant does not automatically ensure a successful long-term outcome. Transplant recipients face challenges with transplant rejection, life-long immunosuppressant therapy and a median survival of 12.5 years in adult patients [8,9]. In addition to the clinical challenges and long wait lists, the high cost of the procedure limits its availability [10].

Using 3D printed patient-specific cardiac models in cardiac therapy has many advantages. Cardiac models that mimic patient anatomy and cardiac tissue composition are used in pre-surgical planning, in simulating complex procedures and in surgical training [11,12]. The cardiac models can be created outside an operating theatre and without invasive procedures for the patient and can contribute to improved patient outcomes with less complications [11]. Three-dimensional-printed models can also feature pathologies and anatomical variation, which can reduce the reliance on using cadaveric material in medical education [13]. In addition, 3D models can be used as a tool to assist in the communication between the patient and healthcare providers. Personalised cardiac models are useful when discussing treatment options and can aid in obtaining an informed consent [12,13]. The advantage of 3D bioprinting (3DBP) is in the ability to use patient-derived stem cells in creating personalised tissue constructs. This allows for individualised treatment tailored to the biochemical profile of the target area, with a decreased risk of transplant rejection [14,15]. Ultimately, 3D-bioprinted cardiac patches may offer an alternative for heart transplants if the method results in cardiac regeneration [16].

Stem cell therapy has shown promise in the treatment of ischaemic heart conditions and HF. Treating the infarcted area with a delivery of patient-derived stem cells has been shown to assist in cardiac regeneration of the damaged tissue [9,10]. However, when injected into the native myocardium, the cells failed due to inadequate cell integration with the host tissue [9]. Cardiac tissue engineering has emerged for its potential to create a biofabricated cardiac patch with cellular material organised within a temporary scaffold structure [17]. Cardiac patches could be delivered to the epicardium for tissue regeneration, and the functionality of the implanted tissue could be tested by measuring ventricular function post implantation [18]. A recent review of small and large animal studies using cardiac patches for treating heart injuries demonstrated improved cardiac function, reduced infarct size and increased angiogenesis [4]. Although promising, all these studies used animal models and results were observed after short follow-up periods, ranging from 1 week to 3 months. This limits the understanding of the safety and efficacy of the treatment in the long term. In addition, only a handful of clinical studies have been conducted to date, which indicates a need for additional clinical trials [19,20,21].

The goal of personalised medicine is to move from standardised healthcare approaches to treatments tailored for individual patients [22]. Designing a personalised cardiac patch should, therefore, be driven by patient specifics, starting with an understanding of the patient’s cardiac anatomy and function. Current literature on cardiac 3DBP tends to focus on biomaterial and bioprinting requirements, spending less time on discussing anatomical considerations or the use of 3D modelling. Defining the design geometry for the personalised cardiac patch is accomplished with the aid of medical imaging data and computer-aided design [23]. Bioprinting cardiac patches to clinically relevant detail and size requires improving the resolution and print speed of current 3D bioprinters [14,24,25]. In addition, increasing the surface area and thickness of the cardiac patch requires branching the vascular tree inside the cardiac patch. This ensures perfusion throughout the engineered tissue [16]. As the majority of current reviews on cardiac 3DBP focus on the processes relevant to biomaterial selection and 3DBP methods, this review excludes the aspect of delivery and engraftment methods. Further information on cell-loaded delivery systems and engraftment techniques can be found in recent publications [26,27]. This review focuses on the design process of a personalised cardiac patch, from obtaining medical imaging data to the bioprinting step. First, it introduces the 3DBP processes used to date, including bioink and 3D design components. Then, it compares findings from current studies to highlight what has been achieved to date and the limitations they present. Finally, it provides recommendations for future studies on how to combine bioprinting processes with anatomically oriented design to improve cardiac patch personalisation.

## 2. Overview

The process of creating a personalised cardiac patch is the sum of several steps requiring methodical planning [17,28,29]. The personalisation process can be divided broadly into cell and biomaterial selection and 3D design, as presented in Figure 1. Before the process starts in a bioengineering laboratory, patients selected for the treatment have their primary cells isolated via a tissue biopsy [30]. The diagram in Figure 1 summarises the following steps and outlines in red areas that use patient-specific input. The section on the left in Figure 1 represents cell line and hydrogel material selection, which results in bioink formulation. This is a critical step in the patient-specific design if the cardiac patch design incorporates patient stem cells. Cell and biomaterial selection also drive biomaterial requirements as the resulting bioink formulation must support cell viability and function [31]. Elements on the right in Figure 1 include steps that capture the patient’s cardiac anatomy from medical imaging. This allows for the defining of the 3D design of the cardiac patch. The design geometry places requirements regarding the mechanical properties of the bioink, which then needs to be compatible with the selected 3DBP method [32]. The success of the cardiac patch transplantation depends on how safely and free of complication the implantation takes place. The post-implantation outcome can be evaluated by measuring the efficacy and viability of the transplanted cells [4,19,30]. The following sections of this review discuss the elements of the two broader areas in Figure 1 as they have been reviewed in current literature.

### 2.1. Bioinks and 3D Bioprinting Methods

In the past two decades, tissue engineering and additive manufacturing (AM) have seen significant technological development and integration of the two fields [27,32,33]. The use of AM in regenerative medicine has led to engineering 3D tissue constructs with biomaterials, which then evolved to the concept of 3DBP [33]. In 2014, Murphy and Atala wrote a landmark review outlining the processes and future challenges in 3DBP [14]. Numerous studies on the use of 3DBP in tissue engineering have been written since their seminal article [34]. Entering the search term “3D bioprinting” AND “review” in Scopus returned 634 results. A further analysis of the search results showed an increase in the number of publications from 1 article in 2012 to 181 in 2021. The section on biomaterial selection in this review analyses 28 3DBP reviews and articles published between 2016 and 2021. A summary of the key topics and recommendations for future research is listed in Table 1 and discussed in more detail in the following sections.

#### 2.1.1. Bioink Formulation

Bioink formulation, or combining cells and biomaterials for 3DBP, forms the biological basis of the patient-specific cardiac patch design. It is also an aspect where the personalisation is at its most critical, as the biological composition impacts the survival rate of the cells and how the cardiac patch can be bioprinted [46].

Biomaterials are biological substances that form a platform for the bioprinted tissue by having the cellular material either seeded on the biomaterial layer or encapsulated within the bioink. They are commonly classified as natural (e.g., alginate, collagen, gelatin, hyaluronic acid), synthetic (e.g., gelatin methacrylate or GelMa, polycaprolactone, polyethylene glycol, pluronics), or hybrid based on the structural fibre [34] or into functional, sacrificial and supportive categories based on their function during the printing process [36]. While there is a large volume of studies on the classes of biomaterials used in 3DBP, [34,35,36,39,42,43,44], a systematic analysis and comparison of their defects (such as printing properties, mechanical properties and biocompatibility) is critical for the optimal biofabrication of bioprinted tissue. For instance, alginate hydrogels have been extensively used in the generation of bioprinted tissues for their high biocompatibility and printability, despite the variability in mechanical properties from one lot to another [54,55]. The addition of collagen and gelatin to alginate hydrogels has provided the required additional bioactivity to promote fundamental cellular behaviours, such as cell–matrix adhesion, proliferation and survival [54,55]. However, given the temperature-sensitive nature of gelatin, the printing process has been challenged by the optimal compromise between cell viability and printability using gelatin hydrogels [54,55]. The development of photoactivated hydrogels, such as GelMa, has opened a new area of biomaterials used in 3DBP for increased printability while maintaining high bioactivity typical of gelatin. Nevertheless, cells differently respond to light exposure during the printing process, leading to the development of a diverse spectra of photoinitiators, such as Irgacure and LAP, for optimal cell viability and function. In addition to the fibre-based biomaterials, tissue spheroids containing cells only have been used as bioinks to create scaffold-free constructs by spontaneous cellular self-assembly. These 3D cell aggregates have been shown to enhance biomimicry. Compared to fibre-based 2D monolayers, tissue spheroids have shown better cellular fusion and diffusion through the bioprinted tissue [4,16,34,35,42,43].

As introduced above, essential biomaterial properties are typically discussed based on their mechanical (printability, rheology, etc.) or biological (biomimicry, biocompatibility, degradation kinetics and by-products) traits [14,15,34,36,44,45]. A different approach to categorising biomaterial characteristics is presented by Hölzl et al. [46]. Instead of focusing on biomaterial parameters, their review discusses bioink properties before, during and after the bioprinting process. Before bioprinting, the importance is placed on cytocompatibility and the ability to encapsulate the cells inside the bioink, with a focus on increasing the cell survivability and promoting maintained cell expression. When bioprinting a biomaterial, this should ideally have good shear thinning properties for high-resolution finish and a fast-cross-linking mechanism to support the layer-by-layer adhesion to stabilise the bioprinted structure. After printing, the swelling behaviour of the biomaterial is an important factor as this has an impact on how the 3D structure achieves its final shape and size. Diffusion of oxygen and nutrients through the bioprinted tissue is critical for cell viability. This mechanism also assists in the removal of waste products from the encapsulated cells.

Cataloguing cell sources and reporting on experiments regarding their suitability for a variety of bioengineered tissues have been covered in a large body of reviews. A selection of recent publications was used in this review to discuss cellular differentiation and maturation. Refer to these for a detailed description of the different cell types and their application [4,15,22,36,40,41,42,47]. Using adult somatic cells isolated from patients is essential for engineering personalised cardiac tissue. Wang et al. [4] and Yadid et al. [40] have recently reviewed the bioengineering of functional cardiac tissues, including methods for differentiating human-induced pluripotent stem cells (hiPSCs) derived from adult somatic cells. HiPSCs can be reprogrammed either into cardiac fibroblasts and cardiomyocytes (CMs) to form muscle tissue or into smooth muscle cells (SMCs) and endothelial cells (ECs) to provide vascularisation within the engineered cardiac tissue. Once reprogrammed, hiPSCs are structurally immature and require mechanical and electrical stimuli to develop a mature phenotype [40,41,42,47]. The maturation process is important for cell survival. The process ensures integration of the engineered tissue with the host myocardium and lowers the risk of arrhythmia after implantation [42]. Similarly, a recently published review of the complexity of cardiac tissue engineering concluded that the maturation process enhances hiPSC-derived CM contractility, force generation and passive stiffness [47]. To enhance the CM function even further, the microenvironment where the cells are matured should mimic the mechanical forces, electrical activity and stiffness of the native myocardium [40,42]. Further strategies that have been suggested to improve the maturation process include using prolonged culture periods, co-culturing CMs with supportive non-CM cell types and maturing the cells in a 3D environment [40,42,47].

#### 2.1.2. Three-Dimensional Bioprinting Methods

The ability to produce engineered tissues using 3DBP is in itself a sign of how much the bioprinting technology has progressed in recent years. Stereolithography, inkjet, extrusion, laser-assisted and electrospinning-based bioprinting strategies have emerged as frontrunners during the past decade [15]. Murphy and Atala [14] discussed the use of inkjet, microextrusion and laser-assisted bioprinting strategies. More recently, the technique and technology of these bioprinting strategies have been covered in numerous publications [34,35,36,39,43,44,45,48,49,50], including two extensive systematic reviews [15,32]. Bioprinters need to adhere to technical specifications to meet the resolution, dimensional accuracy and printing speed requirements for creating cardiac patches. The most commonly cited limitation of currently available bioprinting strategies is the relationship with print resolution and print speed. A fast bioprinting method may not generate products that meet the required accuracy, whereas a high-resolution print may take too long for creating tissue in clinically relevant sizes.

Matching the complexity and diversity of organ and tissue architecture with the right bioprinting method is a balancing act between biocompatibility and printability [30]. Selecting an optimal 3DBP modality depends on the engineered tissue type and the intended use of the bioprinted product [45]. Several current reviews have compared bioprinter capabilities and advantages and disadvantages of different 3DBP modalities currently used in tissue engineering [29,33,39,41,48]. A shared recommendation of the studies is to use extrusion bioprinting for complex scaffold architecture and for continuous, high-density deposition of cells. In contrast, a low rate of cell viability caused by shear stress inside the extruder nozzle was found to be a compromising factor of the extrusion method. One option to reduce the shear stress is using microfluidic bioprinting. The microfluidic method is an emerging 3DBP method which enables a more precise control of the bioink flow and allows extruding more than one type of bioink simultaneously [56]. By using this method, cellular material is extruded through a laminar core surrounded by a sheath flow of one or more biomaterials [56]. This protects the cells from shear stress inside the nozzle during printing and increases cell viability [56,57].

### 2.2. Three-Dimensional Modelling and Design

Creating a 3D digital model from medical imaging data has a well-established role in personalised medicine. Patient-specific 3D models are used in hospitals and medical device manufacturing to tailor approaches to healthcare and treatment planning [48,58]. The same design knowledge from personalised 3D modelling is used in regenerative medicine and 3DBP. Murphy and Atala [14] emphasise the role of medical imaging and anatomical 3D modelling as an essential first step of the 3DBP process. They discuss the use of CT and MRI data for capturing the 3D structure and functionality of tissues and organs and obtaining accurate measurements of the anatomical features. The authors describe the use of computer-aided and mathematical modelling in visualising organ anatomy and predicting the mechanical and biochemical characteristics of bioengineered tissue. Recent reviews and articles on the 3DBP process have generally focused on cell and biomaterial selection and describing the use and biocompatibility of 3DBP methods. Of the 28 current publications listed in Table 1, only three discuss the use of medical imaging and 3D modelling as an integral part of the 3DBP process [30,33,35]. Six articles either mention or recommend the use of patient-specific 3D modelling [22,36,37,38,39,40], and another six limit the discussion to describing cardiac anatomy and biophysical properties [6,22,24,30,33,41].

Creating an accurate, patient-specific cardiac 3D model requires knowledge of the patient’s cardiac anatomy and changes to the ventricular geometry during a cardiac cycle. The data are captured by cardiac imaging, and more than one modality can be used to record the morphological and functional details. Chest X-rays, echocardiograms, cardiac computed tomography (CT) scans, or cardiac magnetic resonance imaging (MRI) or a combination of these is commonly used to analyse cardiac morphology and function [59,60,61]. The volumetric data from CT and MRI scans are captured in adjoining slices, and a continuous 3D structure can be created using fixed image datapoints. This makes the CT and MRI methods ideal for 3D modelling purposes [62]. In addition, the high-level contrast and spatial resolution achieved with CT and MRI scans make the techniques optimal for 3D modelling [63]. Echocardiography is a non-invasive, low-cost and widely accessible cardiac imaging method; however, there are not enough studies on 3D printing cardiac models based on the modality [13]. As a result, the use of echocardiography is excluded from the scope of this review.

Cardiac CT scans are typically used for imaging the morphology of the coronary arteries and anatomical features of the heart [64]. In comparison, cardiac MRI scans capture the soft tissue in greater detail and provide higher-temporal-resolution data compared to cardiac CTs [64]. Perfusion to the myocardium and strain in the ventricular wall can be seen in both scans. However, cardiac MRI is better suited to analyse the cardiac function in relation to ventricular motion [61,65]. The use of cardiac CT and MRI scans and hybrid imaging is discussed further in the following sections. This is followed by sections on 3D modelling and cardiac patch design, addressed from the perspective of personalised anatomical design.

#### 2.2.1. Cardiac CT

Cardiac CT scan is a commonly requested diagnostic test when a detailed and high-contrast visualisation of the cardiac anatomy is required [64,66]. It has a wide range of use in analysing cardiac function, including diagnosing CAD, investigating congenital heart conditions and analysing blood volume and myocardial perfusion [67,68].

As the cardiac CT procedure is non-invasive, it does not pose the risk of arterial damage associated with coronary angiography and it is usually well tolerated by patients [66]. It is usually ECG gated and collects data from a particular part of the cardiac cycle, usually around diastole. Other procedural advantages include short, one breath-hold scan time; capturing the whole left ventricle in one shot; and producing a volume-rendered whole heart model from the scan data [62,64,67]. A limitation of cardiac CT scans is the associated ionising radiation, as this is a risk factor that must be considered when planning the diagnostic approach [24,66,67]. To visualise the cardiac chambers and the arterial network in more detail, an intravenous contrast agent is administered. The contrast agent can be nephrotoxic and carries a risk of contrast allergy and cannot therefore be used in patients with contraindication to iodinated contrast [66,67]. For these patients, a cardiac MRI may be more suitable as it can be performed without a contrast agent. Regardless of the chosen imaging pathway, a risk-benefit analysis should be carried out to determine the best outcome for the patient [66].

Since a cardiac CT can provide a high-definition 3D visualisation of the cardiac anatomy, it meets the fundamental requirement of obtaining accurate geometric data of the ventricular wall [24,62,69]. However, obtaining high-definition, spatially accurate motion data is also essential for designing a patient-specific cardiac patch, and this can be achieved using a cardiac MRI scan.

#### 2.2.2. Cardiac MRI

Cardiac MRI scans detail specifics of the cardiac anatomy and function. Where cardiac CT provides precisely detailed anatomical data, cardiac MRI is used to determine regions of disease in the myocardium and to capture high-definition motion data [61,64,65].

Since cardiac MRI does not use ionising radiation and as most functional cardiac MRI studies can be performed without an intravenous contrast agent, the imaging method is an alternative for patients unable to undergo cardiac CT. It is, however, ruled out for patients with metallic implants or foreign bodies incompatible with MRI [64] and may exclude patients that experience anxiety or claustrophobia when positioned inside the MRI machine [24,70]. Regions of disease, myocardial perfusion and fibrotic scarring in the myocardium can be identified by using a contrast agent [71]. Measurements of cardiac function, including stroke volume and ejection fraction, assess the efficacy of the cardiac patch before and after transplantation. For further details on the methods of analysing cardiac function with cardiac MRI, we recommend an excellent review by Peng et al. [71]

In comparison to the fast scan times of cardiac CTs, cardiac MRIs have a longer acquisition time as they capture 2D data 1–2 slices at a time [72]. During the MRI scan, patients are required to sustain multiple breath-holds to reduce respiratory motion artefact. This may rule out the imaging method for younger or sick patients [72]. If the breath-holding amplitude is inconsistent, neighbouring slices can get misregistered and produce errors in the volumetric data [60,72]. State-of-the-art MRI scanners and software using compressed sensing can potentially capture the entire area in all cardiac phases and in a single breath-hold [73].

Cardiac cine MRI is a protocol used to measure global cardiac function [64]. Cine MRI data gated to the ECG is acquired by imaging the same region of interest several times at defined time points. When applied in cardiac imaging, consecutive frames are taken over corresponding time points in the cardiac cycle to produce a movie of the ventricular movement [71,72]. If precise localised motion data are required, cardiac MRI tagging can be used for analysing regional myocardial deformation [71,74,75]. The MRI tagging method is based on altering the net magnetisation of a specified region in the patient myocardial wall [60]. The localised alteration creates a contrast with adjacent untagged areas, which can be seen as a grid pattern in the MRI scans. As the tag pattern follows ventricular motion, changes to the local geometry during a contractile cycle are recorded [65,71].

The diagnostic information relating to the areas of MI is essential for the personalisation process. This will determine the shape, size and dynamic and mechanical characteristics of the patient-specific cardiac patch. High-definition motion data are equally important for the cardiac patch personalisation, as it is vital to understand the dynamic changes to the region of interest during a cardiac cycle. The cardiac patch must adjust to the changing shape of the ventricular wall, withstand the forces of the changing forces and amplitude and remain positioned in a correct location and orientation on the epicardium.

#### 2.2.3. Hybrid Imaging

Although cardiac CT and cardiac MRI scans provide accurate information for diagnostic purposes, hybrid imaging can produce a more informative visualisation compared to individual scans [61,76]. Hybrid imaging involves data acquisition typically from two imaging modalities. Segmentation of the scan data, co-registration using place markers, fusion of the imaging data and volume rendering will result in a 3D model [77,78]. The 3D model created using hybrid data will indicate myocardial regions with disrupted perfusion caused by coronary occlusions. The end result is a personalised 3D model of the patient coronary artery morphology and ventricular function.

A pilot study of the hybrid imaging method investigated the correlation of coronary artery stenoses and myocardial defects [77]. The study combined cardiac CT and cardiac perfusion MRI data and resulted in a precise 3D visualisation of the area of myocardial infarction caused by coronary occlusions. A more recent paper compared diagnostic results from separate cardiac CT and cardiac perfusion MRI scans to a 3D fusion image from the same data [78]. Although the study was limited by a small cohort size, the results were promising and suggest that the hybrid method can give more accurate diagnostic guidance compared to individual studies.

#### 2.2.4. Three-Dimensional Modelling

Three-dimensional modelling of the cardiac CT and cardiac MRI data is a step where the precision of the imaging method becomes crucial. Inaccuracies in the data can impact the final shape and fidelity of the 3D model [58] and impact the patient-specific cardiac patch design.

Once the cardiac imaging data have been obtained, they are segmented using specialised software. Commonly used segmentation methods, such as thresholding and region growing, have been reviewed and described previously [23,79]. High-resolution data from a cardiac CT scan are optimal for 3D modelling and relatively easy and fast to segment [62]. In a newly published study, 3D printed models created from segmented CT scan data were scanned and the surface scan data were compared to originating STL-files for alignment and registration [80]. The study validated the accuracy of using segmented CT scan data for producing 3D models, as results showed outstanding accuracy with an average error of 7 µm. Although the accuracy of MRI data is considered lower than that of data from CT scans, current reviews point out the difficulty of comparing the data accuracy between the two methods [58,71]. A common finding in the current reviews is the time-consuming nature of the manual and semi-automated segmentation processes and the requirement for using dedicated software [23,79].

After the segmentation process, a stereolithography (STL) file is generated from the segmented scan data. The STL format breaks the geometry into a mesh formed out of triangle vertices that follow the contours of the model [62]. If a physical representation of the 3D model is required, the STL-file typically requires post-processing before it can be sent for 3D printing [80,81]. This may include removing artefacts, smoothing surfaces, filling holes and gaps and cropping the model to a region of interest [11,62].

#### 2.2.5. Cardiac Patch Design

There are several crucial considerations for the design geometry of the personalised cardiac patch. It must contour to the local dimensions and anatomy as defined by the cardiac imaging; it must have the correct mechanical parameters of stiffness, elasticity and rigidity to suit the ventricular motion; and it needs to facilitate the bioink requirements for building a scaffold to provide the structural framework [27]. Essentially, the design geometry should be based on the patient-specific cardiac 3D model to match the patient anatomy.

Current studies have been found to have a mixed approach to defining dimensions for cardiac patches. Since in vitro studies do not involve transplantation, their tendency is to focus on optimising the engineered tissue for bench testing and not consider the overall geometric design [17,82]. The overall shape and size of the bioprinted implant is a central design aspect, especially in large animal and clinical studies. Larger tissue implants require vascularisation and innervation within the design to ensure long-term cell survival and prevention of necrosis [83]. This has been shown to be difficult to achieve with the modern 3D bioprinters as current print resolution does not enable bioprinting channels < 100 μm in diameter [83]. Using hydrogels in bioprinting enables creating soft and flexible designs. However, their disadvantage is poor long-term stability and material degradation [24]. The gel-like consistency of hydrogels requires a supporting scaffold during bioprinting due to material deformation [17,24,25]. Apart from the bioprinting stage, this may present a challenge during the transport and engraftment if the large and complex 3D geometry does not hold shape. A scaffold-free method of printing to a supporting hydrogel bath could be used to prevent material deformation during bioprinting [17,24,25]. Supporting the bioprinted 3D cardiac patch after the sacrificial material has been removed should be planned so that the implant maintains shape and integrity till the engraftment stage.

Tissue-engineered cardiac patches have been engrafted to the myocardial wall in several in vivo studies, including in small and large animal models [4,41,84], but significantly also in a small number of clinical studies on humans [19,20,21]. In general, small animal studies did not discuss specific, morphology-based designs for the cardiac patch. Parameters for the engineered tissue tend to focus on ensuring that the mechanical characteristics match the native myocardium, the size is fit for the biofabrication process and the small animal, or that the design geometry can withstand the delivery method and surgical engraftment [85,86,87].

In two large animal studies, a surgically induced defect was covered with a correspondingly sized patch without reliance on cardiac imaging for shaping the transplant [88,89]. Although the methods included using multilayered sheet material for the cardiac patch generation, the findings are significant as they demonstrated the performance of the cardiac patch in vivo in an MRI [88] and visualised in electrical and optical mapping [89]. The same studies also had maximum follow-up periods, of 12 and 8 months, respectively, indicating the viability of cardiac patches for up to 1 year from implantation.

Some clinical studies have used multilayered cell sheets rather than bioprinted cardiac patches to treat patients with ischemic heart disease [19,20] or dilated cardiomyopathy [21]. Patches were transplanted on to the myocardial wall of the left ventricle, and cardiac imaging was used to measure cardiac function pre- and postoperatively. All studies describe methods for creating the multicellular sheets and in one case mention the cardiac patch size [19]. None, however, discuss shaping the patch or using imaging studies in the design process. It should also be noted that current literature has not been found to consider the design geometry in relation to the patient-specific cardiac anatomy or functional characteristics. This is a distinct gap in the current literature and will be addressed in the discussion.

## 3. Discussion

Is it possible to bioprint a personalised cardiac patch for treating heart failure? In theory, and based on the emerging studies discussed in our review, yes, but areas requiring further investigation remain. As discussed in this review, studies to date confirm that we can capture the cardiac anatomy and function to a detail that allows producing an accurate 3D model of the area of myocardial infarction. A personalised cardiac patch can be designed from the resulting cardiac 3D model in a shape and a size to match the infarcted region. Software and hardware are available for segmenting the data from medical imaging and converting the 3D model to a format suitable for 3DBP. Technology for the 3DBP process is used in cardiac tissue engineering today, and novel methods that can combine several biomaterials are currently being developed.

Using a bioink where the cells and biomaterial have been derived from the patient’s own stem cells is a benchmark for creating a cardiac patch with a personalised biological profile. This technique is already a reality, as a fully personalised cardiac patch using patient stem cells and decellularised ECM (dECM) for biomaterial was 3D bioprinted recently [17]. Although this is an important finding for improved biomimicry, a question remains as to whether an allogeneic hydrogel is always the best option or whether would another type of material perform as well or better. Synthetic biomaterials may be easier to source and may provide other benefits, such as increased mechanical strength.

Finding optimal parameters for the overall design may also require a compromise between the bioink composition and the 3DBP method. Print requirements vs. cellular requirements for bioinks may require compromising depending on the type of cells used and the application of the tissue. A potential solution may be to combine cell-optimised hydrogels with other biomaterials, such as nanorods, to achieve mechanical stability and shape fidelity [46]. Despite the rapid development of 3D bioprinters, the call remains for increased 3DBP resolution for producing finer details, increased printing speed for ability to scale up printed tissue size and the creation of multi-axial and multi-arm printers [15,30,34,38,39,44,51]. To produce a 3D-bioprinted organ requires advancing from bioprinting uniform tissue to complex, heterogenous tissue architecture [24]. Achieving a heterogeneous tissue anatomy means bioprinting nano- and microscale structures within one tissue construct [24]. This increases the need for higher print resolution even further and underlines the importance of developing currently available bioprinters and 3DBP methods. Novel 3DBP technologies and methods are emerging at the same time. Of particular interest are the development of in situ and 4D bioprinting methods [6,14,15,22,32,34,36,38,39,41,50,52]. Nevertheless, further considerations and studies focusing on biomaterials with optimal mechanical and printing properties and biocompatibility are essential for the success of the bioprinting process.

Although cardiac MRI scans can be used for a variety of diagnostic studies, the lengthy duration of the scan rules it out for patients unable to sustain breath-holding. Three-dimensional cine steady-state free precession (SSFP) imaging uses a method where the image acquisition time is considerably shorter and a data set can be captured in a single breath hold [72]. It may be that as cardiac MRI protocols, especially the 3D cine MRI, develop further, adequate anatomical data can be obtained using one modality. This would reduce the number and type of scans required for the design process, saving in healthcare cost, but, more importantly, requiring the patient to have only one scan and without exposure to ionising radiation.

As discussed in sections regarding cardiac imaging, a personalised cardiac 3D model visualises patient-specific anatomy and function of the heart. Current literature appears to have a gap in this regard, as current discussion tends to focus on a singular aspect: either creating an accurate representation of the cardiac anatomy from CT scans or mapping the cardiac biophysical data from cardiac MRIs. The need for specialist knowledge across different fields of science and medicine in 3DBP is nonetheless recognised. Collaborative studies between healthcare professionals, imaging and 3D modelling specialists, designers and engineers have been recommended for creating accurate cardiac 3D models [12,90,91].

The process of cardiac image segmentation can be made faster and less labour intensive by using mathematical and computational modelling. Computational 3D cardiac models enable automation of the segmentation process and simulation of cardiac physics, as presented in comprehensive reviews published earlier [69,91]. According to the reviews, computational cardiac models allow for fast and accurate modelling of cardiac anatomy and enable creating highly specific cardiac biophysical simulations. The downside is the complexity of the approach and specialist skills required to use the software, which may slow the adoption rate of the technology [91].

The method for creating a personalised cardiac patch to fit the patient anatomy and changes to the cardiac wall has not been tested to date. As a result, it is unclear how accurately the cardiac patch design needs to match the native anatomy to be fit for purpose. The bioink may be yielding enough not to require perfect accuracy for the patch fit, and the print resolution may not need to be of the highest quality if a product 3D-bioprinted using lower resolution is adequate. The region of interest in the left ventricle and a proposal for the personalised cardiac patch design obtained from a cardiac CT scan and cardiac MRI data are presented in Figure 2.

Current literature mainly discusses post-operative complications associated with the engraftment rate and transplant integration with the host myocardium [27,93]. The personalised cardiac patch design in Figure 2c represents a concept of a large cardiac patch that matches the region of interest on the patient myocardium. When transplanted, the cardiac patch is orientated to the bi-pedal human anatomy, which differs from the quadrupedal animal models in relation to diaphragmic movement and intrathoracic pressure. The pericardium is attached to the central tendon of the diaphragm via the phrenopericardial ligament [94]. The movement of the diaphragm causes fluctuation in the intrathoracic pressure. This can be sudden and forceful during deep inspiration and expulsion in coughing [95,96]. It is important to establish post-operatively that the cardiac patch stays in its intended position. Due to the small number of existing clinical studies, it is unknown if the diaphragmic movement impacts the stability of the cardiac patch placement. This would require future studies to make sure the cardiac patch remains in its intended location and also further research on optimal engraftment techniques to restrict displacement.

An important cost consideration is avoiding over-engineering the design and production method, as this will increase the financial cost and production time. Optimising the design by excluding zones and volume inessential for the cardiac patch functionality should also be considered, as this will reduce material use. A cost–benefit analysis could similarly be carried out by testing a series of cardiac patches, ranging from fast printing, low-resolution patches to highly engineered and detailed designs. This may help to enhance the design and lower material use if the patch thickness and printer requirements can be adjusted to meet specific requirements.

In addition to design and material considerations, studies used in this review called for standardisation of biofabrication practices, better quality control and adherence to good manufacturing practices [16,22,34,39,41,44,45,50,51]. Due to the complexity and wide range of application of 3D bioprinting, standardising the process may not, however, be practical. Instead, to improve the reproducibility of studies and to ease comparing findings, developing a minimum information guideline for 3DBP projects and establishing applicable parameters for different organs is recommended [83]. Adding to this, some of these studies discussed the impact of ethical and regulatory matters on the 3DBP process.

A vast body of studies has been published analysing and categorising biomaterials, cellular sources and 3DBP methods. The same observation was made on the number of reviews on the use of patient-specific 3D design and 3D modelling in medicine. Future studies should extend their scope to include aspects from both areas and review the well-covered topics when new studies come out. Finally, as addressed at the start of this review, if the personalised cardiac patch can be harnessed successfully for use in cardiac tissue regeneration, this method could be finally explored as a much-needed solution for patients currently on a long waiting list for a heart transplant.

## Figures and Tables

**Figure 1 bioengineering-09-00093-f001:**
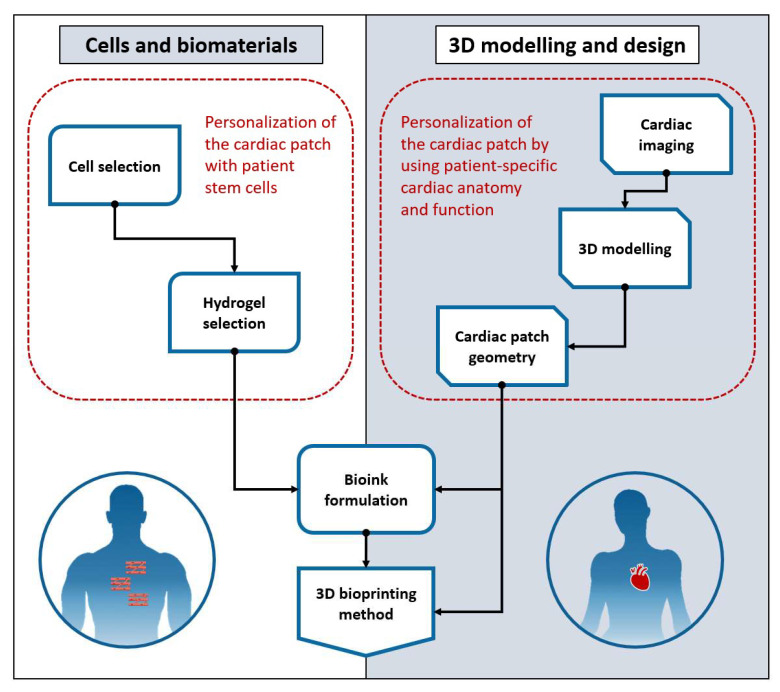
Biological and 3D design considerations of creating a personalised cardiac patch; the red border indicates components that can be matched closest to patient specifics. Selection of cells and hydrogels is a critical design component as the use of patient stem cells forms a biological basis for creating a personalised patch. The resulting bioink formulation must support cell viability and function. Anatomical and functional characteristics of the heart are captured from cardiac imaging data and used to create a patient-specific 3D cardiac model. The design geometry of the cardiac patch is based on the personalised cardiac 3D model. The shape, the size and the thickness of the patch impact the bioink formulation and the selection of the 3DBP method used.

**Figure 2 bioengineering-09-00093-f002:**
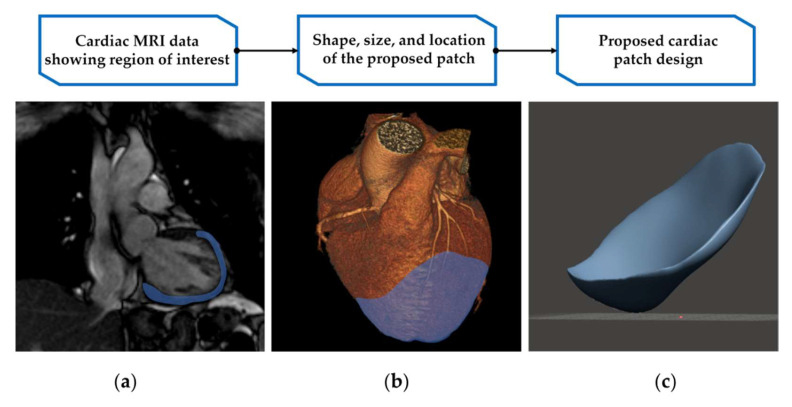
Using cardiac MRI and cardiac CT data for creating a 3D presentation of the cardiac patch model: (**a**) cardiac MRI viewed on the short axis plane, a region of interest on the left ventricle and the apical area highlighted in blue (DICOM image sample sets from [92]); (**b**) anatomy of the whole heart is captured from a cardiac CT scan with the region of interest highlighted in blue (DICOM image sample sets from [92]; (**c**) a proposal for a 3D model of a cardiac patch based on a digital model created from cardiac imaging data.

**Table 1 bioengineering-09-00093-t001:** Summary of important factors and challenges in 3D bioprinting as reported by Murphy and Atala in 2014 [14] and as discussed in current reviews and articles.

Factor	Murphy and Atala, 2014 [14]	Current Literature	Refs.
3D modelling and design	Presents medical imaging and 3D anatomical modelling as an essential first step of the 3DBP process	Discusses medical imaging and 3D modelling as part of the 3DBP process	[30,33,35]
	Discusses use of CT and MRI data for obtaining tissue dimension measurements	Mentions or recommends the use of patient-specific 3D modelling	[22,36,37,38,39,40]
	Presents the use of computer-aided and mathematical modelling for obtaining and digitising anatomical features	Describes cardiac anatomy and/or biophysical properties	[6,22,24,30,33,41]
Biomaterials	Discusses essential biomaterial properties	Presents biomaterials used in 3DBP	[34,35,36,39,42,43,44]
	States the importance of an optimal scaffold structure for mechanical and functional integrity	Discusses essential biomaterial properties	[14,15,34,36,44,45,46]
Cell sources	Talks about cell selection in generic terms	Reviews cell sources used in 3DBP	[4,15,22,36,40,41,42,47]
	Addresses cellular proliferation and cell requirements for tolerating mechanical and biological stress	Lists requirements for control of cell maturation	[40,41,42,47]
3D bioprinting methods	Description of inkjet, microextrusion and laser-assisted 3DBP strategies	Description of 3DBP technology and strategies	[15,32,34,35,36,39,43,44,45,48,49,50]
	Comparison of bioprinter capabilities	Comparison of bioprinter capabilities	[32,34,36,44,51]
Challenges and future research	Increasing 3DBP printer resolution, printing speed and compatibility with current and future biomaterials	Increasing 3DBP printer resolution, printing speed and creating multi-axial/multi-arm printers	[15,30,34,38,39,44,51]
	In vivo bioprinting of cells and materials directly on or in the patient	Emergence of new hybrid, in situ and 4D bioprinting methods	[6,15,22,32,34,36,38,39,41,50,52]
	Producing a map of ECM protein structures and distribution within an organ	Developing cell types and novel biomaterials towards better biomimicry	[6,15,22,30,32,34,35,36,39,41,42,44,48,50,52]
	Developing complex, hybrid, or functionally adaptive biomaterials for improved 3DBP compatibility	Vascularisation, contractility and maturation	[6,16,30,33,34,35,36,39,40,42,45,47,51,52,53]
	Combining different cell types in one tissue; control of proliferation and differentiation	Ethical and regulatory considerations, standardisation of biofabrication practices and quality control	[16,22,34,39,41,44,45,50,51]
	Vascularisation, innervation and maturation of the bioprinted tissue		

## Data Availability

Not applicable.

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
