# Peer review of "Taking It Personally: 3D Bioprinting a Patient-Specific Cardiac Patch for the Treatment of Heart Failure"

_bioengineering, 2022, doi:10.3390/bioengineering9030093_

Round 1

Reviewer 1 Report

This mini-review mainly covered two aspects relevant to the design of a personalized cardiac patch. It described bioink and 3D design components and discussed the use of medical imaging and 3D modelling as an integral part of cardiac 3D bioprinting (3DBP) Finally, it provides recommendations for future studies on how to combine bioprinting processes with anatomically oriented design to improve cardiac patch personalization. The paper can be used for reference in designing the cardiac patch and is probably publishable after revision.

  1. The necessity and advantages of 3D printing in cardiac therapy should be described in the introduction.
  2. Although the author introduced biomaterials as components of bioink in 2.1.1 part and table 1, there was little information of the materials in the part that have been used in 3DBP and let alone systematic analysis and comparison their defects (such as printing properties, mechanical properties and biocompatibility), which are essential for the part and should be described in more detail.
  3. Figure 2 showed the authors’ proposal for the personalized cardiac patch design obtained from cardiac CT scan and cardiac MRI data, how to obtain the patch, the bioprinting method and possible bioinks that could be used for the patch should be further proposed.
  4. There are many complex sentence structures in the paper, please thoroughly revised them for better understanding.

Reviewer 2 Report

The presented review paper is related to bioprinting a patient-specific cardiac patch. Achieving the physiological relevancy of brioprinted tissues is the current problem of bioengineering. The authors reviewed a significant part of updated literature to 2021.

Some of the recommendations:

1. The source of Figure #2 needs to be cited. If Figure #2 is the original figure, it should be removed (!) or completed by Materials and methods, Ethical committee approval, and Written informed consent.

2. The review pays little attention to the postoperative complications of 3D printed cardiac patches implantation. The paper needs to be updated by a Complication-related subsection.

3. In the paper, more attention should be paid to the Design problems of 3D-printed implants. I recommend using these references: 
1. Arguchinskaya, N. V., Beketov, E. E., Kisel, A. A., Isaeva, E. V., Osidak, E. O., Domogatsky, S. P., ... & Kaprin, A. D. (2021). The Technique of Thyroid Cartilage Scaffold Support Formation for Extrusion-Based Bioprinting. International Journal of Bioprinting, 7(2), 348.
2. Tian, S., Zhao, H., & Lewinski, N. (2021). Key parameters and applications of extrusion-based bioprinting. Bioprinting, e00156.

Round 2

Reviewer 1 Report

I think the authors have solved all concerns.